# Vitamin D Status and Indices of Mineral Homeostasis in the Population: Differences Between 25-Hydroxyvitamin D and 1,25-Dihydroxyvitamin D

**DOI:** 10.3390/nu11081777

**Published:** 2019-08-01

**Authors:** Massimo Cirillo, Giancarlo Bilancio, Ermanno Guarino, Pierpaolo Cavallo, Cinzia Lombardi, Simona Costanzo, Amalia De Curtis, Augusto Di Castelnuovo, Licia Iacoviello

**Affiliations:** 1Department of Public Health, University of Naples “Federico II”, 80131 Naples (NA), Italy; 2Department of Medicine, Surgery and Odontoiatry “Scuola Medica Salernitana”, University of Salerno, 84081 Baronissi (SA), Italy; 3Department of Physics, University of Salerno, 84084 Fisciano (SA), Italy; 4Istituto Sistemi Complessi, Centro Nazionale Ricerche, 00185 Rome, Italy; 5Maternal-Infant Department, Hospital “*San Pio*”, 82028 Benevento (BN), Italy; 6Departement of Epidemiology and Prevention, IRCCS Neuromed, 86077 Pozzilli (IS), Italy; 7Mediterranea Cardiocentro, 80133 Napoli (NA), Italy; 8Department of Medicine and Surgery, Research Center in Epidemiology and Preventive Medicine (EPIMED), University of Insubria, 21100 Varese (VA), Italy

**Keywords:** 25-hydroxyvitamin D (calcidiol), 1,25-dihydroxyvitamin D (calcitriol), parathyroid hormone, calcium, phosphorus, epidemiology

## Abstract

Opinions are conflicting about the epidemiology of vitamin D deficiency. This population-based study investigated cross-sectionally the associations of 25-hydroxyvitamin D (calcidiol) and 1,25-dihydroxyvitamin D (calcitriol) with indices of mineral homeostasis. Study cohort consisted of 979 persons of the Moli-Sani study, both sexes, ages ≥35 years. Data collection included serum calcidiol by different assays, serum calcitriol, serum parathyroid hormone, serum and urine calcium, and phosphorus. Prevalence of mild-to-moderate calcidiol deficiency (10–19 ng/mL) was 36.4% and did not associate with hypocalcemia or hyperparathyroidism. Prevalence of severe calcidiol deficiency (<10 ng/mL) was 16.8% and associated with hyperparathyroidism only (odds ratio = 8.81, 95% confidence interval = 2.4/32.9). Prevalence of calcitriol deficiency (<18 pg/mL) was 3.1% and associated with hypocalcemia (29.1, 7.4/114.5) but not hyperparathyroidism. In ANOVA along concentration strata, lower calcidiol associated with higher parathyroid hormone only (*p* < 0.001). Lower calcitriol associated with lower serum and urine calcium (*p* < 0.001) but not with parathyroid hormone. Calcidiol findings were consistent with different calcidiol assays. In the population, mild-to-moderate calcidiol deficiency did not associate with abnormal mineral homeostasis. Severe calcidiol deficiency and calcitriol deficiency associated with different disorders: lower calcidiol associated with hyperparathyroidism whereas lower calcitriol associated with hypocalcemia and low urine calcium.

## 1. Introduction

Vitamin D plays a key role in calcium homeostasis [1]. For this effect, 25-hydroxyvitamin D (calcidiol) is regarded as a precursor with low or no biologic activity, whereas 1,25-dihydroxyvitamin D (calcitriol) is considered the most active modulator [1]. In contrast with this view, high serum calcidiol is considered per se a risk factor for hypercalciuria [2]. Several authors have reported a high prevalence of vitamin D deficiency in the population because of the evidence of serum calcidiol below 20 ng/mL [3,4,5]. Others have argued that this was a misinterpretation of the concept of vitamin D deficiency and that serum calcidiol <20 ng/mL rarely implies true vitamin D deficiency [1,6]. Additional uncertainties in the definition of vitamin D status are due to the confounding effects of factors such as the variable accuracy of vitamin D assays, the influence of vitamin-D binding protein, and albumin on serum calcidiol levels [7,8,9].

Hypocalcemia and high parathyroid hormone (PTH) are regarded as classical marks of hypovitaminosis D [10,11], but their associations with vitamin D deficiency have never been investigated in the general population. Epidemiologic studies found an inverse association of serum calcidiol with serum PTH but did not report data on high PTH, serum calcium, or hypocalcemia [12,13,14]. Therefore, the present study investigated, in a sample of the adult general population, the associations of serum calcidiol and serum calcitriol with hypocalcemia, high PTH, and other indices of mineral homeostasis.

## 2. Material and Methods

### 2.1. Study Design and Population

The Moli-sani study is a prospective cohort study ongoing since 2005 that enrolled 24,325 persons, both sexes, ages ≥35 years, randomly recruited from the general population of a region of central-southern Italy [15]. The study complies with the World Medical Association Declaration of Helsinki–Ethical Principles for Medical Research Involving Human Subjects and was approved by the Rome Catholic University ethical committee (P99, A.931/03-138-04—11 February 2014). All participants provided written informed consent.

The main purpose of the Moli-sani study is to investigate risk factors for degenerative diseases. During the recruitment visit, questionnaires were administered for information about socioeconomic status, physical activity, medical history, dietary habits, risk factors for cardiovascular disease and/or tumor, and family medical history. The medical examination included measurements of blood pressure, anthropometry, spirometry, and standard electrocardiogram. Venous blood samples were obtained after an overnight fast. Untimed urine spot samples were collected from first void at wake up. Biological samples were processed within 3 h and stored in liquid nitrogen as described [16]. Baseline lab tests for the whole cohort included the measurements of serum calcidiol and cystatin C as part of the BiomarCaRE project [17].

The present paper reports an observational, cross-sectional study in a sub-cohort of the Moli-sani study. The sub-cohort was selected to have 100 men and 100 women for each of the five-following age-strata: 35–44, 45–54, 55–64, 65–74, and ≥75 years. For this sub-cohort, frozen samples were used for additional lab tests that included a repeated measurement of serum calcidiol with the use of a different assay and new measurements for serum concentrations of calcitriol, PTH, albumin, total calcium, phosphorus, and creatinine, and for urine concentrations of calcium, phosphorus, and creatinine.

### 2.2. Lab Procedures

The measurements of serum calcidiol for the BiomarCaRE project were performed by immunoassay [17]. The repeated measurement of calcidiol and the new measurement of calcitriol were performed by a fully-automated chemiluminescent assay (Diasorin, Saluggia, Italy) [18]. Variability of chemiluminescent method in blind duplicates was <5% within- and between-assay (Appendix A). Other information in the Appendix A shows that the results of the assays were stable over the time required to complete the measurements in the whole sub-cohort (Appendix A). The calibration of the calcidiol chemiluminescent assay was evaluated using NIST-SRM 972a [19,20]. This product contains two certified levels of calcidiol, as assessed by concordant results of gold standard methods, that is of isotope dilution liquid mass chromatography (ID-LC-MS) and of isotope dilution liquid chromatography-tandem mass spectrophotometry (ID-LC-MS/MS) [21,22]. With the use of the Diasorin calibration, NIST-SRM 972a averaged −2.1% of the certified level 18.9 ng/mL, and −12.0% of the certified level 33.2 ng/mL. Thus, the chemiluminescent calcidiol assay was re-calibrated using a quadratic equation that best-fitted the non-linear relation between measured levels and NIST-certified levels (Appendix A). Therefore, the analysis dealt with three sets of calcidiol data: The immunoassay data of the BiomarCaRE project (defined as immunoassay data), the chemiluminescent assay data (non-recalibrated data), and the chemiluminescent assay data re-calibrated with use of NIST-SRM 972a (recalibrated data). Results for immunoassay data and non-recalibrated data are reported in the Appendix A.

The other new measurements were performed by automated biochemistry (Abbott, Chicago, Illinois, IL, USA) [23]. Serum creatinine was measured by enzymatic method calibrated with IDMS-traceable standard [24]. Variability in blind duplicates was <5% for all these measurements.

Serum and urine calcium, serum and urine phosphorus, and serum PTH were used as indices of mineral homeostasis. Serum calcium was analyzed without normalization, and with normalization for serum albumin [10]. To reduce the confounding of errors in timing and completeness of urine collection, urine calcium and urine phosphorus were evaluated as calcium/creatinine ratio and phosphorus/creatinine ratio, respectively [25,26]. Body mass index (BMI = weight_kg_/height_m_^2^) was used as index of overweight. Kidney function was assessed as estimated glomerular filtration rate (eGFR), calculated by the combined creatinine-cystatin C equation of the Chronic Kidney Disease Epidemiology Collaboration study [27,28].

### 2.3. Outcome Variables and Statistical Analysis

First, the analysis investigated the association of standard definitions of vitamin D deficiency with hypocalcemia and high PTH, which are considered disorders typical of hypovitaminosis D [29]. Calcidiol deficiency was defined severe when serum calcidiol <10 ng/mL, and mild-to-moderate when serum calcidiol 10–19 ng/mL [2]. Calcitriol deficiency was defined when serum calcitriol <18 ng/mL [30]. Hypocalcemia was defined when serum calcium ≤8.6 mg/100 mL [31], and high PTH when serum PTH ≥66 pg/mL [32]. Second, the analysis examined the relationships of the whole range of serum calcidiol and serum calcitriol with indices of mineral homeostasis.

Statistical procedures were performed using IBM-SPSS 19 and included correlation and regression analysis, paired *t*-test, ANOVA, ANOVA with control for sex, age, BMI and eGFR (other variables), chi-square analysis, area under the curve of receiver operating characteristic curve (ROC_AUC_), and McNemar test for comparison of paired categorical data. ANOVA was used to investigate the statistical significance of contrasts or of linear trends. To control for double testing, the Bonferroni correction was used and *p*-values were considered significant when <0.025 (0.05/2).

## 3. Results

### 3.1. Descriptive Statistics

The study cohort consisted of 979 persons with complete data (51.2% men; age = 59.9 ± 9.8 years). Serum calcidiol distribution was positively skewed in the recalibrated dataset (Figure 1), as well as in the other two datasets (Appendix A). Data of different datasets of serum calcidiol highly correlated with each other but averaged differently (Appendix A). Appendix A shows other descriptive statistics.

### 3.2. Association of Vitamin D Deficiency with Hypocalcemia and High PTH

Using the recalibrated dataset, prevalence was 16.8% for severe calcidiol deficiency and 36.5% for mild-to-moderate calcidiol deficiency (*n* = 164 and 357, respectively). Severe calcidiol deficiency associated with high PTH but not with hypocalcemia (Table 1). Mild-to-moderate calcidiol deficiency did not associate with hypocalcemia or high PTH. Mild-to-moderate deficiency was more prevalent using other calcidiol datasets (Appendix A). Findings for calcidiol deficiency and hypocalcemia or high PTH were similar using other datasets (Appendix A).

Prevalence of calcitriol deficiency was 3.1% in the whole cohort. Calcitriol deficiency was higher in persons with calcidiol deficiency compared to persons without calcidiol deficiency (*n* = 521 and 458, prevalence = 4.4% and 1.5%, *p* = 0.009) but was similar between mild-to-moderate and severe deficiency of calcidiol (4.5% and 4.3%, *p* = 0.912). Calcitriol deficiency associated with hypocalcemia but not with high PTH (Table 1).

### 3.3. Indices of Mineral Homeostasis over the Range of Serum Calcidiol and of Serum Calcitriol

For analyses over the whole range of serum calcidiol and serum calcitriol, the study cohort was divided into strata as shown in Figure 1. The stratum with the highest calcidiol level (≥70 ng/mL, *n* = 3) and the stratum with the lowest calcitriol level (<10 pg/mL, *n* = 3) were combined with the respective next stratum to avoid the bias of very low *n*. Serum and urine calcium did not differ along serum calcidiol strata, whereas they were linearly higher along serum calcitriol strata (Figure 2). Findings were similar with log-transformation of calcidiol or calcitriol data (not shown), or normalizing serum calcium for serum albumin (Appendix A) or using other calcidiol datasets (Appendix A). With control for other variables (sex, age, BMI, and eGFR), the association of serum calcidiol with serum calcium turned significantly negative, whereas other findings did not vary (Appendix A).

Serum PTH was linearly lower along serum calcidiol strata and not significantly different along calcitriol strata (Figure 3). Findings were similar with log-transformation of calcidiol or calcitriol data (not shown), or using other calcidiol datasets or with control for other variables (Appendix A).

Serum phosphorus was linearly higher along serum calcidiol strata but weakly lower along serum calcitriol strata (upper panels of Figure 4). Urine phosphorus insignificantly varied along calcidiol and calcitriol strata (lower panels of Figure 4). Findings were similar with log-transformation of calcidiol or calcitriol data (not shown) or using other calcidiol datasets or with control for other variables (Appendix A).

## 4. Discussion

In comparison to previous epidemiologic studies, the present results confirmed the high prevalence in the general population of serum calcidiol in the range currently defined as mild-to-moderate deficiency [2,3,4,5]. Calcidiol deficiency in the Moli-sani cohort, compared to a Mediterranean population residing in Spain [14], was more prevalent either using the threshold of 10 ng/mL (prevalence = 17% and 5%, respectively) or the threshold of 20 ng/mL (53% and 40%, respectively). Present results confirmed also the inverse association of serum calcidiol with serum PTH [12,13,14]. Other comparisons with previous studies are impossible because of the lack of previous population-based data on serum or urine levels of calcium or phosphorus.

The main limitations of the study were the low number of persons with calcitriol deficiency and the lack of data on serum ionized calcium, timed urine collections, other phosphoro-tropic molecules (e.g., fibroblast growth factor 23), bone densitometry, persons with age <35 years, and other ethnic groups. The low number of persons with calcitriol deficiency widened the confidence intervals of the association with hypocalcemia. The lack of data on serum ionized calcium was partially compensated by analyses with albumin-normalized serum calcium [10]. The use of untimed urine spot sample limited the generalization of results to 24-h urine collections, but reduced errors in timing and completeness inevitable for urine collection at the level of the general population. Other phosphoro-tropic molecules are rarely included in the clinical diagnostic work-up and are never investigated at the epidemiological level. Lack of information on younger adults and different ethnic groups could not be solved and could limit the generalizability of the findings. Merits of the study were the analysis of different calcidiol assays [7], the recalibration of calcidiol assay with gold-standard reference [19,20,21,22], the first assessment of serum calcitriol in the general population, the inclusion in the analysis of serum and urine levels of calcium and phosphorus, and the accurate eGFR calculation.

With regard to calcidiol, the lack of positive associations of serum calcidiol with serum or urine calcium was in perfect accordance with the lack of effects of vitamin D supplementation on serum or urine calcium in a randomized controlled trial in hypertensives [33], supporting the view of negligible direct effects of calcidiol on calcium homeostasis. Vice versa, the independent association of lower serum calcidiol with higher serum PTH was in perfect accordance with the in vitro and in vivo inhibition of calcidiol on parathyroid hormone secretion [33,34,35]. Considering that PTH lowers serum phosphorus, the evidence of lower serum phosphorus in persons with lower serum calcidiol could reasonably reflect a secondary consequence of the association between lower serum calcidiol and higher PTH, rather than a direct effect of calcidiol on phosphorus homeostasis. A similar reasoning could apply for the seemingly paradoxical association of lower serum calcidiol with higher serum calcium that appeared after control for sex, age, BMI, and eGFR.

With regard to calcitriol, the positive linear association of calcitriol with serum calcium was in accordance with the concept that this form of vitamin D plays a direct key role in calcium homeostasis, increasing intestinal calcium absorption and, thus, serum calcium [1,29]. The positive linear association of calcitriol with urine calcium further supported this interpretation. The lack of a strong association of serum calcitriol with serum PTH seemed in contrast with the in vitro inhibition of calcitriol on PTH secretion [34]. However, given that PTH stimulates calcitriol generation [1,29], the study results could actually reflect the combination of two opposite effects: A calcitriol-dependent inhibition of PTH secretion [34], in combination with a PTH-dependent stimulation of calcitriol generation [36]. In a similar way, the weak inverse association of serum calcitriol with serum phosphorus could be the result of the combination between the PTH-dependent increase in calcitriol generation and the PTH-dependent decrease in serum phosphorus. Last, the lack of associations of serum calcitriol with urine phosphorus indicates that calcitriol unlikely affected intestinal phosphorus absorption, at least in the presence of the phosphorus intake present in this population and within the observed range of serum calcitriol concentrations.

With regard to calcidiol/calcitriol inter-relationships, the study results suggest that factors other than calcidiol status play a major role in the development of calcitriol deficiency. Calcitriol deficiency was, in fact, similarly prevalent in calcidiol deficiency of different severity and was at times detectable in the absence of any calcidiol deficiency.

Altogether, the results suggest a model where the classic markers of hypovitaminosis D are often dissociated. Per se, calcidiol deficiency associated only with a secondary increase in PTH secretion and with predictable PTH-dependent effects on serum phosphorus. Signs of calcium deficit, such as low serum calcium up to clinically defined hypercalcemia and/or low urine calcium, lacked in calcidiol deficiency and appeared only when calcitriol deficiency was present. Although it was more frequent in the presence of calcidiol deficiency, calcitriol deficiency was certainly ascribable also to other determinants of calcitriol generation such as PTH levels and kidney function [36,37].

Practical implications of the study were that, for mineral homeostasis, mild-to-moderate calcidiol deficiency rarely represents a reliable evidence of hypovitaminosis D given that the vast majority of persons with serum calcidiol in that range was without clinically significant alterations in serum calcium, serum PTH, and serum calcitriol. The results suggest that a panel of lab tests, including serum and urine calcium, serum PTH, serum phosphorus, and serum calcitriol, improved the detection and the definition of disorders characterized by the presence or the risk of hypovitaminosis D. The results suggest also that high serum PTH in the general population may, at times, represent a disorder secondary to low calcidiol levels. Last, the results did not support the idea that serum calcitriol in the range 50–80 ng/mL might increase the risk of hypercalciuria [2].

## 5. Conclusions

The study reported the first population-based results about serum levels of calcidiol and calcitriol, together with lab indices of mineral homeostasis. Mild-to-moderate calcidiol deficiency did not associate with sizeable disorders in mineral homeostasis. Severe calcidiol deficiency associated with secondary relative hyperparathyroidism but not with signs of calcium deficiency. Reductions in serum and/or urine calcium were observed only in the presence of calcitriol deficiency. Prevalence of calcitriol deficiency was <5% even among persons with severe calcidiol deficiency.

## Figures and Tables

**Figure 1 nutrients-11-01777-f001:**
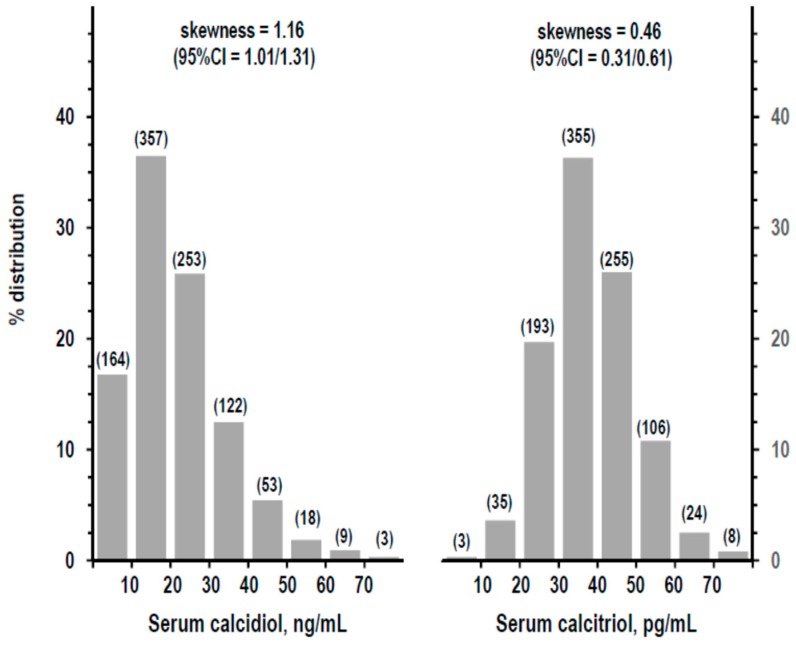
Frequency distribution and skewness (95% CI) of serum calcidiol (left panel, recalibrated data) and of serum calcitriol (right panel). The number of persons per stratum of concentration is shown within parentheses on the top of each bar.

**Figure 2 nutrients-11-01777-f002:**
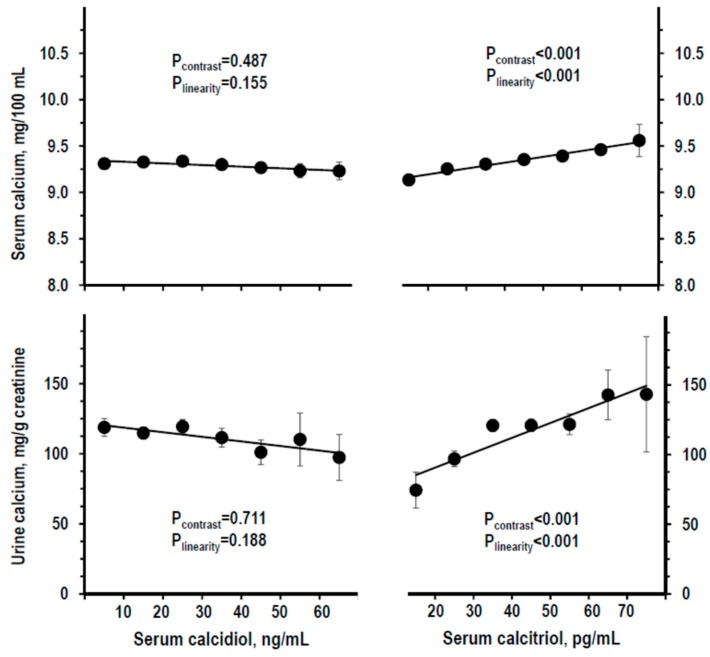
Mean ± SE of serum calcium (top panels) and of urine calcium/creatinine ratio (lower panels) by stratum of serum calcidiol (left panels, recalibrated data) and of serum calcitriol (right panels). Strata were defined as in Figure 1. The stratum with serum calcidiol ≥70 ng/mL and the stratum with serum calcitriol <10 pg/mL were combined with the next stratum to avoid the bias due to low *n* (*n* = 3). Number of persons per stratum from left to right: For calcidiol = 164, 357, 253, 122, 53, 18, and 12; for calcitriol = 38, 193, 355, 255, 106, 24, and 8. *p*-values by non-adjusted ANOVA. *p*-values were considered statistically significant when <0.025. The line shows the linear trend along strata.

**Figure 3 nutrients-11-01777-f003:**
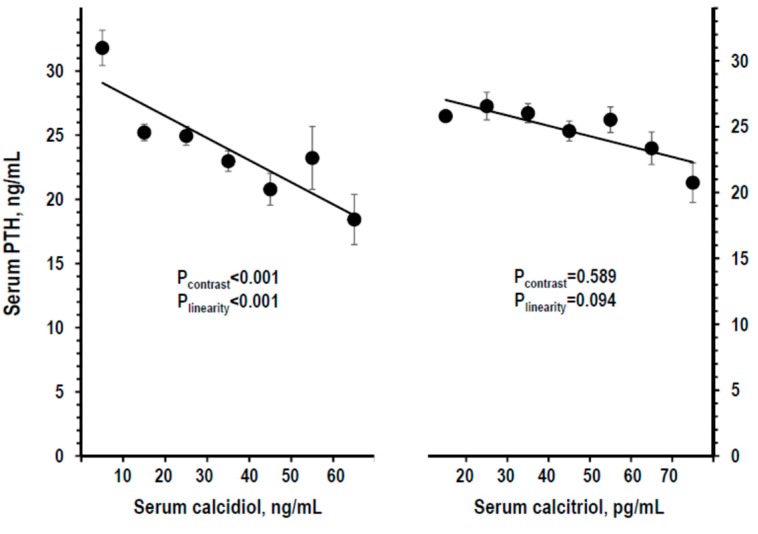
Mean ± SE of serum PTH by stratum of serum calcidiol and of serum calcitriol (left and right panel, respectively). Strata were defined as in Figure 1. The stratum with serum calcidiol ≥70 ng/mL and the stratum with serum calcitriol <10 pg/mL were combined with the next stratum to avoid the bias due to low *n* (*n* = 3). Number of persons per stratum from left to right: For calcidiol = 164, 357, 253, 122, 53, 18, and 12; for calcitriol = 38, 193, 355, 255, 106, 24, and 8. *p*-values by non-adjusted ANOVA. *p*-values were considered statistically significant when <0.025. The line shows the linear trend along strata.

**Figure 4 nutrients-11-01777-f004:**
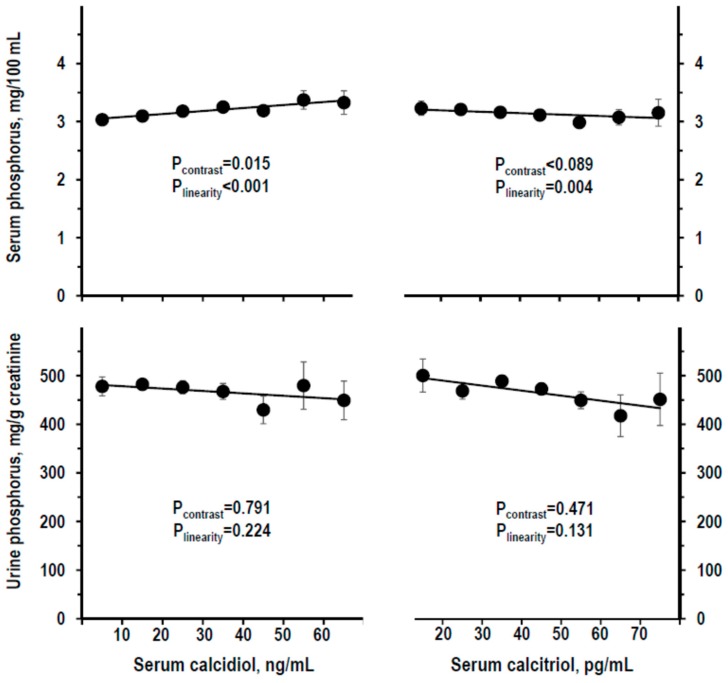
Mean ± SE of serum phosphorus (top panels) and of urine phosphorus/creatinine ratio (lower panels) by stratum of serum calcidiol (recalibrated data, left panels) and of serum calcitriol (right panels). Strata were defined as in Figure 1. The stratum with serum calcidiol ≥70 ng/mL and the stratum with serum calcitriol <10 pg/mL were combined with the next stratum to avoid the bias due to low *n* (*n* = 3). Number of persons per stratum from left to right: For calcidiol = 164, 357, 253, 122, 53, 18, and 12; for calcitriol = 38, 193, 355, 255, 106, 24, and 8. *p*-values by non-adjusted ANOVA. *p*-values were considered statistically significant when <0.025. The line shows the linear trend along strata.

**Table 1 nutrients-11-01777-t001:** Association of deficiency of calcidiol (data of the recalibrated chemiluminescent assay) with hypocalcemia and high serum parathyroid hormone (PTH): Prevalence, odds ratio, and area under the curve of receiver operating characteristic curve (ROC_AUC_) with 95%CI.

	Hypocalcemia ^a^			High Serum PTH ^b^		
	Prevalence	Odds Ratio *(95%CI)*	ROC_AUC_ *(95%CI)*	Prevalence	Odds Ratio *(95%CI)*	ROC_AUC_ *(95%CI)*
**Serum calcidiol**						
with severe deficiency <10 ng/mL *n* = 164	1.8%	1.69 *(0.40/7.14)*	0.556^ns^ *(0.35/0.76)*	5.5%	8.81 *(2.35/32.9)*	0.748 ** *(0.61/0.89)*
with mild-to-moderate deficiency 10–19 ng/mL *n* = 357	0.3%	0.25 *(0.03/2.19)*	0.363^ns^ *(0.17/0.56)*	1.1%	1.72 *(0.38/7.73)*	0.567 ^ns^ *(0.35/0.78)*
without deficiency ≥20 ng/mL *n* = 458	1.1%	1 (ref)		0.7%	1 (ref)	
**Serum calcitriol**						
with deficiency <18 pg/mL *n* = 30	13.3%	29.05 *(7.37/114.5)*	0.709 * *(0.50/0.92)*	0.0%	0.97 *(0.96/0.98)*	0.484 ^ns^ *(0.34/0.62)*
without deficiency ≥18 pg/mL *n* = 949	0.5%	1 (ref)		1.7%	1 (ref)	

^a^ hypocalcemia = serum calcium ≤8.6 mg/100 mL; ^b^ high PTH = serum PTH ≥66 pg/mL; Significance of ROC_AUC_: ^ns^ not significant, * *p* = 0.031, ** *p* = 0.003.

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
