# Peer review of "Vitamin D Status and Indices of Mineral Homeostasis in the Population: Differences Between 25-Hydroxyvitamin D and 1,25-Dihydroxyvitamin D"

_nutrients, 2019, doi:10.3390/nu11081777_

Round 1

Reviewer 1 Report

Thank you for inviting me to review this paper. It is reasonably well written and contains a lot of information.

Major comments:

I am not an expert in the measurement of vitamin D metabolites (or of laboratory methods in general) so I cannot comment on the technicalities. However, the authors do seem to have put effort into calibrating their data and providing detail on the assays conducted (e.g. duplicates, calibration etc).

I am not enough of an expert in this specific area of vitamin D to be able to fully assess the novelty of the work, or the authors’ claims that this is the first population based results of this kind (L267).  I suspect that it is not a highly novel concept (there are lots of published papers on calcidiol, calcitriol, serum calcium, PTH) but it may be novel in that these markers have not been analysed in this particular way before.

For the each of the ANOVAs, two analyses have been done (adjusted and unadjusted) so the authors need to control for multiple testing. It is probably easiest to use Bonferroni (i.e. use P=0.05/2 as the cut-off for statistical significance as you have performed two tests for each set of variables). In Figures 2-4 as well as figures S4-S10 (in the Supplementary file)- the P value cut-off should be adjusted. The paper will need to be updated in response to any change in statistical significance.

The y axes of the figures should start from y=0, in order to provide less distortion of the data. By ‘zooming in’ like this it makes the results look artificially stronger.

Minor comments:

L125 I assume the reason for the data being put into strata was because of this skewed calcidiol data? This could be made clearer to the reader. Was log transformation tried first to see if this would normalise the data? (I know this doesn’t always work…).

L167 ‘Insignificantly’ should be ‘not significantly’ as it is not usual/technically correct to use the term ‘statistical insignificance’.

Table 1: The number of people with deficiency of calcitriol is very small (n=30_ so this estimate may be less reliable (and may explain the wide confidence intervals). This could be mentioned in the discussion (limitations).

Author Response

responses are attached in the word fle

Reviewer 2 Report

Major comments:

This is a very interesting study, addressing important questions, the manuscript is good to read and the message clear. However, the novelty of the findings is limited. Additionally, this is a cross-sectional study, not an intervention study. The association of calcidiol and PTH is well-known and not novel.

The authors state the following:

“Other comparisons with previous studies are impossible because of the lack of previous  population-based data on serum or urine levels of calcium or phosphorus.”

Even if this may be correct, I believe that indeed comparisons with other studies can and should be drawn and that the data derived from this study should be put into context with previous publications. Even if study populations are not directly comparable, it might be important to note associations of vitamin D levels (and supplementation) with, for instance, levels of calcium in other populations. For example, in an RCT in hypertensive, but otherwise healthy, adults vitamin D supplementation had no influence on serum and urinary calcium (Pilz S. et al., Effects of Vitamin D on Blood Pressure and Cardiovascular Risk Factors, Hypertension 2015). I think such findings should be discussed. 

Also, please compare your prevalence data to other studies. 

Minor comments:

-      Please remove the results from the discussion section.

Author Response

responses are attached in the word file
